# Nickel-catalysed migratory hydroalkynylation and enantioselective hydroalkynylation of olefins with bromoalkynes

Xiaoli Jiang[1], Bo Han[1], Yuhang Xue[1], Mei Duan[1], Zhuofan Gui[1], You Wang [1✉] & Shaolin Zhu [1✉]

α-Chiral alkyne is a key structural element of many bioactive compounds, chemical probes, and functional materials, and is a valuable synthon in organic synthesis. Here we report a NiH-catalysed reductive migratory hydroalkynylation of olefins with bromoalkynes that delivers the corresponding benzylic alkynylation products in high yields with excellent regioselectivities. Catalytic enantioselective hydroalkynylation of styrenes has also been realized using a simple chiral PyrOx ligand. The obtained enantioenriched benzylic alkynes are versatile synthetic intermediates and can be readily transformed into synthetically useful chiral synthons.

[1] State Key Laboratory of Coordination Chemistry, Jiangsu Key Laboratory of Advanced Organic Materials, Chemistry and Biomedicine Innovation Center (ChemBIC), School of Chemistry and Chemical Engineering, Nanjing University, Nanjing, China. ✉email: wangyou@nju.edu.cn; shaolinzhu@nju.edu.cn

A s a key structural element, chiral alkynes motifs bearing an α stereocentre are often found in many bioactive compounds, chemical probes, and functional materials (Fig. 1a). In addition, they are also valuable synthons as the

$sp^3$-hybridized carbons could undergo versatile transformations to deliver useful $sp^2$- or $sp^3$-hybridized carbons[1]. As a result, efficient strategies for catalytic, enantioselective $C(sp^3)$–$C(sp)$ coupling to generate such stereocentres have long been sought

**Fig. 1 Ni(I)H-catalyzed migratory hydroalkynylation and enantioselective hydroalkynylation. a** Representative bioactive molecules bearing a chiral alkyne motif. **b** Common strategies for $C(sp^3)$–$C(sp)$ coupling. **c** Chemo- & stereoselective NiH-catalyzed (migratory) hydroalkynylation of alkenes.

(Fig. 1b). For example, Liu[2,3] reported an elegant work on Cu-catalyzed asymmetric Sonogashira C($sp^3$)–C($sp$) coupling[4,5]. Shi[6] and Liu[7] have demonstrated that Pd- and Cu-catalyzed C($sp^3$)–H alkynylation could be achieved in an enantioselective fashion. Liu[8] has also used an alkene difunctionalization strategy to produce enantioenriched alkynylation product under copper catalyst[9]. Suginome[10] has reported a pioneering work on Ni-catalyzed asymmetric hydroalkynylation of 1,3-dienes based on their previous hydroalkynylation works[11,12]. As a continued development of general alternatives for asymmetric C($sp^3$)–C($sp$) coupling, here we report an appealing approach via metal-hydride[13–15] catalyzed asymmetric (remote) hydroalkynylation[16] from readily available alkene starting materials.

Owing to its low-cost, facile oxidative addition, and availability of diverse oxidation states, nickel[17,18] has emerged as a catalyst complementary to palladium over the past two decades, especially in cross-coupling reaction involving C($sp^3$) fragments. Reductive migratory hydrofunctionalization[19–22] catalyzed by nickel hydride[23–25] has recently been recognized as an alternative protocol for selective functionalization of remote C($sp^3$)–H bonds[26–66]. Compared to conventional cross-coupling, this process (i) employs readily available, bench-stable alkenes or alkene precursors instead of specially generated organometallic reagents as starting materials and (ii) could also selectively functionalize a remote C($sp^3$)–H site in addition to the conventional *ipso*-position. Since its conception, significant progress has been made toward this synthetically useful process[26–61], which requires that the cross-coupling partner (e.g., aryl halide or alkyl halide) could selectively capture an alkylnickel species generated through iterative migratory insertion/β-hydride elimination.

To explore this nickel-catalyzed migratory hydrofunctionalization further, we recently investigated if a bromoalkyne, an unsaturated C($sp$) cross-coupling partner which is potentially reactive toward NiH, could be used to achieve remote hydroalkynylation (Fig. 1c, i). Successful implementation of this transformation will require (i) a hydrometalation process that can discriminate between alkene and alkyne and (ii) an alkynylation process highly selective for one of the alkylnickel species. A chiral alkyne bearing an α-aryl-substituted stereogenic C($sp^3$) center[2–5,7–10,67,68] would be ultimately obtained from styrene through hydronickellation and subsequent enantioconvergent[52,53,55,56,69] alkynylation (Fig. 1c, ii). Here, we show the successful execution of this reaction.

## Results

**Reaction design and optimization.** Our initial studies involved the migratory hydroalkynylation of 4-phenyl-1-butene (**1a**) using 1-bromo-2-(triisopropylsilyl)acetylene (**2a**) as an alkynylation reagent (Fig. 2). It was determined that NiI$_2$·xH$_2$O and the bathocuproine ligand (**L**) could generate the desired migratory alkynylation product as a single regioisomer [rr (benzylic product: all other isomers) > 99:1] in 82% yield (entry 1). Other nickel sources such as NiBr$_2$ led to lower yields and a moderate rr (entry 2). Ligand screening revealed that the previously used ligand[29], 6,6′-dimethyl-2,2′-bipyridine (**L1**) resulted in significantly lower yield and rr (entry 3) while a similar ligand neocuproine (**L2**) led to a similar regioselectivity but a lower yield (entry 4). Other silanes such as trimethoxysilane and diethoxymethylsilane gave diminished yields (entries 5 and 6), and marginally lower yield was obtained when reducing the amount of PMHS to 2.5 equiv (entry 7). K$_3$PO$_4$·H$_2$O was shown to be an unsuitable base (entry 8). The addition of NaI as an additive improves both the yield and rr, presumably by promoting the regeneration of NiH species (entry 9). An evaluation of solvents

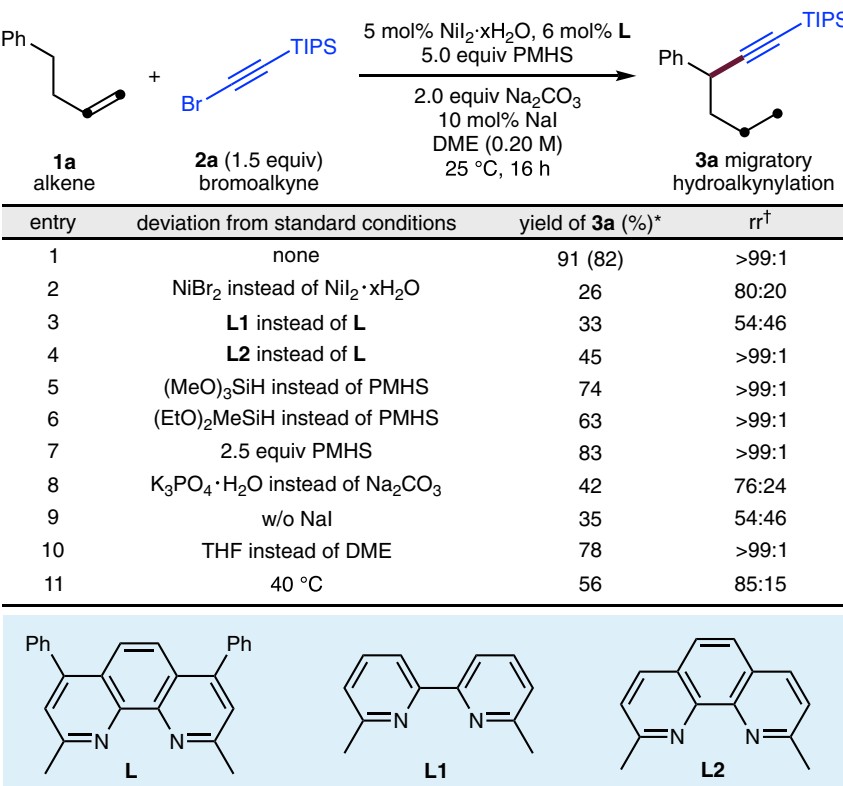

| entry | deviation from standard conditions | yield of **3a** (%)* | rr† |
|-------|-----------------------------------|----------------------|-----|
| 1 | none | 91 (82) | >99:1 |
| 2 | NiBr$_2$ instead of NiI$_2$·xH$_2$O | 26 | 80:20 |
| 3 | **L1** instead of **L** | 33 | 54:46 |
| 4 | **L2** instead of **L** | 45 | >99:1 |
| 5 | (MeO)$_3$SiH instead of PMHS | 74 | >99:1 |
| 6 | (EtO)$_2$MeSiH instead of PMHS | 63 | >99:1 |
| 7 | 2.5 equiv PMHS | 83 | >99:1 |
| 8 | K$_3$PO$_4$·H$_2$O instead of Na$_2$CO$_3$ | 42 | 76:24 |
| 9 | w/o NaI | 35 | 54:46 |
| 10 | THF instead of DME | 78 | >99:1 |
| 11 | 40 °C | 56 | 85:15 |

**Fig. 2 Variation of reaction parameters.** *Yields determined by GC using *n*-dodecane as the internal standard, the yield in parentheses is the isolated yield. †rr refers to regioisomeric ratio, representing the ratio of the major product to the sum of all other isomers as determined by GC analysis. PMHS polymethylhydrosiloxane, DME dimethoxyethane, TIPS triisopropylsilyl.

revealed that THF was less effective than DME (entry 10), and conducting the reaction at 40 °C gave inferior results (entry 11).

**Substrate scope.** With these optimal reaction conditions, we examined the generality of the reaction. As shown in Fig. 3a, unactivated terminal alkenes bearing electron-donating (**3c**) or electron-withdrawing (**3d–3g**) substituents on the remote aryl ring are tolerated. A variety of functional groups are readily accommodated, including ethers (**3c**, **3h–3k**, **3m**), a trifluoromethyl group (**3d**), and esters (**3g**, **3i**). Importantly, tosylates (**3j**) and triflate (**3k**) commonly used for further cross-coupling, all remained intact. The reaction could also proceed with olefin substrate having longer chain length between the starting C=C bond and the remote aryl group, producing the benzylic alkynylation product exclusively although with a lower yield (**3l**). Remarkably, both silyl and sterically hindered alkyl substituted ethynyl bromides work well in this reaction (**3m**, **3n**). Moreover, a variety of unactivated internal alkenes also proved to be competent

coupling partners, regardless of the *E/Z* configuration or the position of the C=C bond (Fig. 3b, **3o–3w**). As expected, styrenes themselves smoothly undergo hydroalkynylation to produce the benzylic alkynylation product exclusively (Fig. 3c, **3x–3k'**). Under these exceptionally mild reaction conditions, various substituents on the aryl ring (**3z–3e'**) as well as heteroaromatic styrenes (**3f'**, **3g'**) were also suitable for this reaction.

In an effort to obtain enantioenriched benzylic alkynylation products, the asymmetric version of NiH-catalyzed hydroalkynylation of styrenes was explored and the results are in Fig. 4. It was found that a chiral PyrOx ligand (*S*)-**L**\* under modified reaction conditions could produce the desired hydroalkynylation products in good yields and excellent ee. Styrenes with a variety of substituents on the aromatic ring underwent asymmetric hydroalkynylation smoothly (**5a–5q**), including ethers (**5d–5i**), an easily reduced aldehyde (**5l**), a nitrile (**5m**, **5n**), and esters (**5o–5q**). Substituents commonly used for further cross-coupling such as aryl chloride (**5c**), aryl bromide (**5k**), and boronic acid pinacol ester (**5j**) all

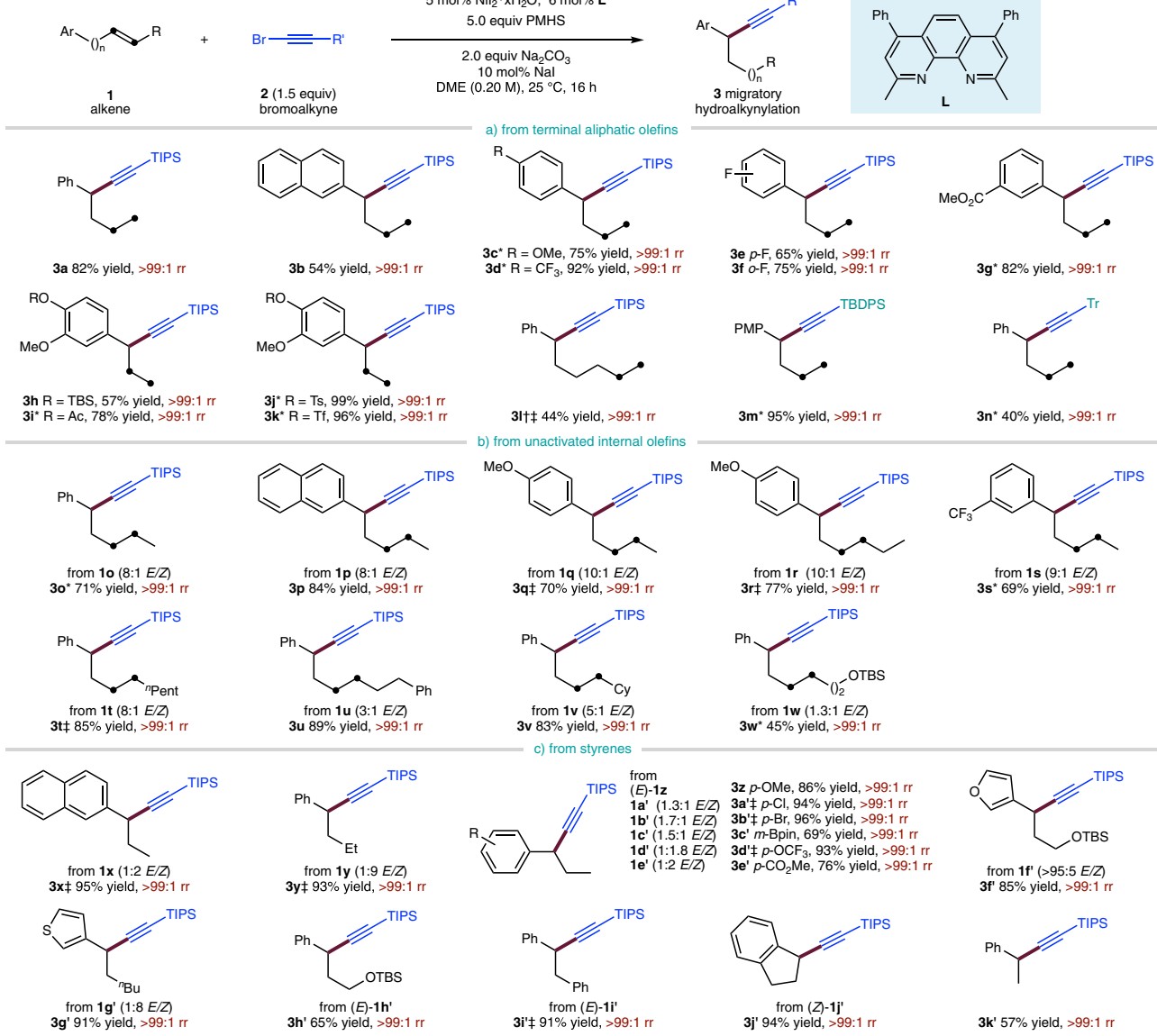

**Fig. 3 NiH-catalyzed migratory hydroalkynylation of alkenes with bromoalkynes.** Yield under each product refers to the isolated yield of purified product (0.20 mmol scale, average of two runs), rr refers to regioisomeric ratio, representing the ratio of the major product to the sum of all other isomers as determined by GC analysis. *Diglyme was used as solvent. †10 mol% NiI₂·xH₂O, 12 mol% **L**, and 20 mol% NaI were used. ‡DME (0.10 M) was used. TBS *tert*-butyldimethylsilyl, TBDPS *tert*-butyldiphenylsilyl; Tr trityl (triphenylmethyl).

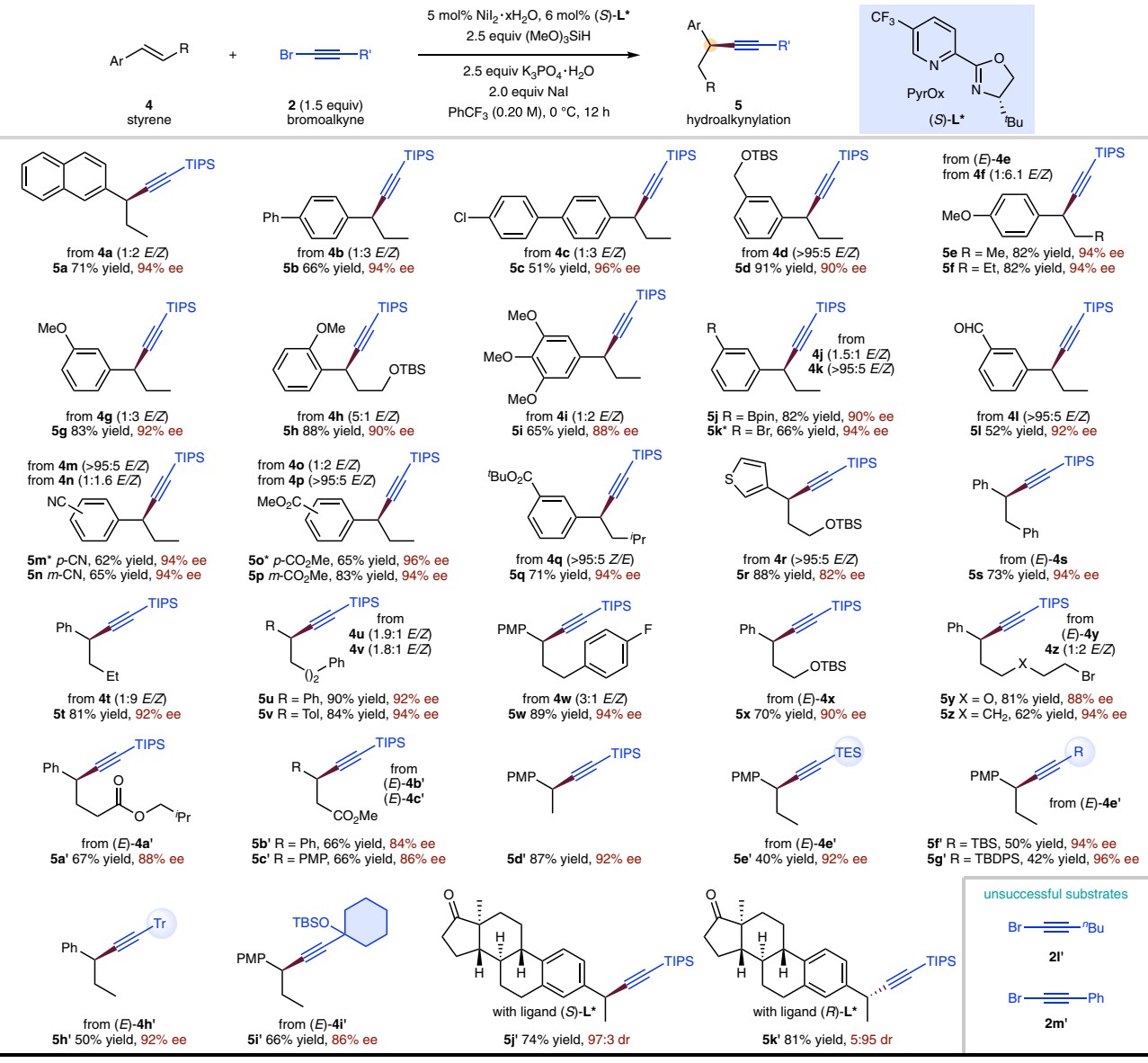

**Fig. 4 NiH-catalyzed enantioselective hydroalkynylation of styrenes with bromoalkynes.** Yield under each product refers to the isolated yield of purified product (0.20 mmol scale, average of two runs), single regioisomer was obtained unless otherwise noted. Enantioselectivities were determined by chiral HPLC analysis. *NiBr₂·diglyme used as catalyst, 1,2-dichloroethane used as solvent, 3.0 equiv NaI used. TES triethylsilyl.

emerged unchanged. The substituents at β-position could also be varied (**5r–5c'**). Alkyl bromides were compatible with the current reaction, providing a synthetic handle for further derivatization (**5y, 5z**). β-Unsubstituted styrenes were also compatible (**5d', 5j', 5k'**). The scope of bromoalkynes was also explored and a range of different sterically hindered substituents at the β-position, including silyl and alkyl-substituted ethynyl bromides were shown to be viable substrates (**5e'–5i'**). However, it should also be noted that the less steric hindered alkyl-substituted ethynyl bromide (**2l'**) and aryl-substituted ethynyl bromide (**2m'**) were unsuccessful substrates[70] and could easily undergo decomposition under the current conditions.

## Discussion

The asymmetric migratory hydroalkynylation could also be realized. In a preliminary experiment with 3-aryl-1-propene (**1i**) as

substrate (Fig. 5a), chain-walking and subsequent asymmetric alkynylation at benzylic position product ((S)-**3i**) was obtained with excellent ee (90% ee) as major isomer (90:10 rr). When the reaction was conducted on a 5 mmol scale, the functionalized chiral benzylic alkyne (**5e**) was obtained in high yield and with excellent enantioselectivity (Fig. 5b). To highlight the synthetic utility of the method, subsequent derivatizations were carried out (Fig. 5c). Desilylation of **5e** yielded the enantioenriched terminal alkyne (**6**), which could further undergo a click reaction to form **7** or a hydration reaction to form **8**. The semi-hydrogenation of alkyne (**5a**) by DIBAL-H (diisobutylaluminum hydride) could be highly stereoselective, giving the Z-alkene (**9**). In addition, oxidative cleavage of the triple bond in **5e** could afford the corresponding chiral carboxylic acid (**10**).

To gain further insight into the mechanism of the hydrometallation process, isotope labeling experiments were conducted. As shown in Fig. 5d, the use of the deuterated trans-alkene (E-**4h-D**)

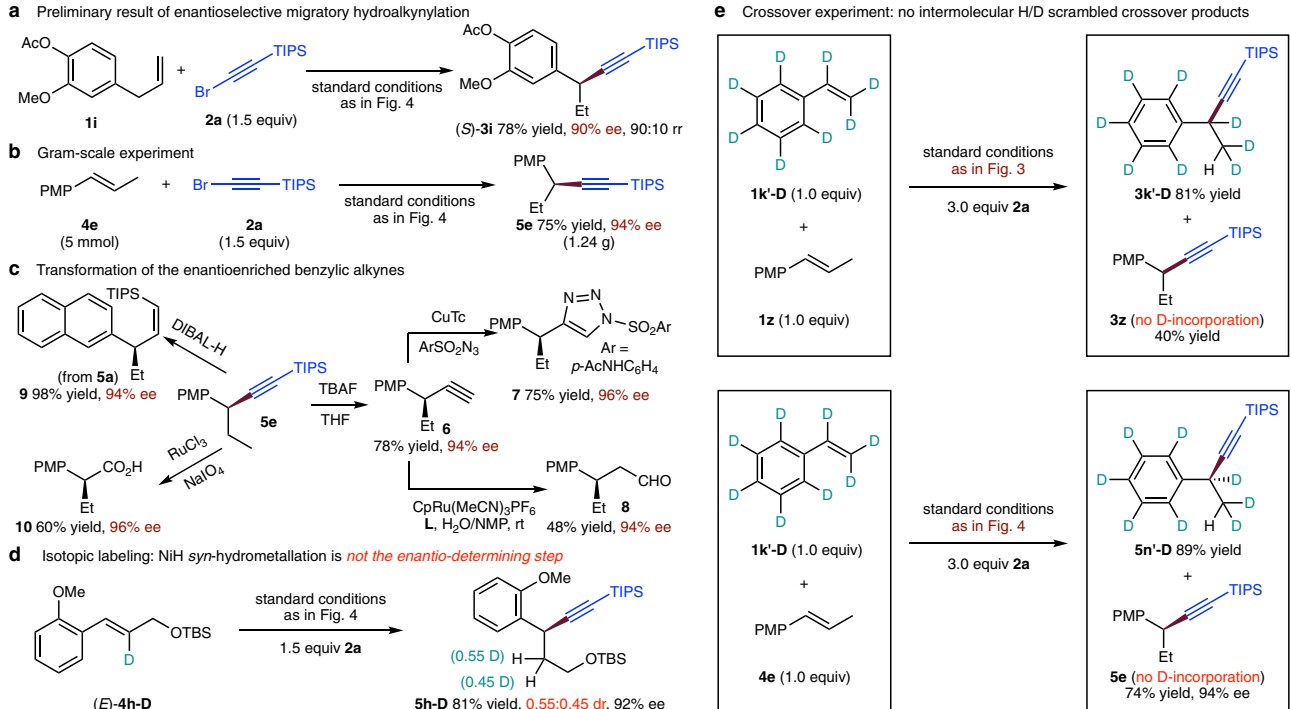

**Fig. 5 Enantioselective migratory hydroalkynylation, gram-scale, derivatization, and isotopic labeling experiments. a** Preliminary result of enantioselective migratory hydroalkynylation. **b** Gram-scale experiment. **c** Transformation of the enantioenriched benzylic alkynes. **d** Isotopic labeling experiment. **e** Crossover experiment.

led to the formation of both diastereomers in approximately equal amounts (0.55:0.45 dr), which indicated that the *syn*-hydrometallation is not the enantio-determining step because if it was, a diastereomerically pure **5h-D** should be formed. This observation is consistent with our initial mechanistic proposal that the benzylic stereocentre is formed through rapid homolysis of the alkyl-Ni(III) bond and subsequently enantioconvergent process, reforming only one Ni(III) enantiomer from Ni(II) and benzylic radical (see Fig. 1c, ii). Furthermore, no intermolecular H/D scrambled crossover products were obtained in both migratory and asymmetric hydroalkynylation conditions, revealing that hydrometallation of NiH/NiD species to styrene is irreversible (Fig. 5e).

In conclusion, we report a NiH-catalyzed strategy to form functionalized benzylic alkynylation products, which are versatile synthetic intermediates. Both migratory hydroalkynylation and asymmetric hydroalkynylation can be realized. These two mild, efficient, and straightforward processes tolerate a wide range of functional groups on both the alkene and bromoalkyne components. A broad substrate scope as well as synthetic utility of this protocol have been demonstrated. An investigation of the mechanism and the development of a migratory enantioselective version of this transformation are currently in progress.

## Methods

**NiH-catalyzed migratory hydroalkynylation of alkenes**. In a nitrogen-filled glove box, to an oven-dried 8 mL screw-cap vial equipped with a magnetic stir bar were added NiI₂·xH₂O (3.8 mg, 5.0 mol%), **L** (4.3 mg, 6.0 mol%), Na₂CO₃ (42.4 mg, 2.0 equiv), NaI (3.0 mg, 10.0 mol%) and anhydrous DME (1.0 mL). The mixture was stirred for 20 min at room temperature (stirred at 800 rpm) before the addition of PMHS (60 μL, 1.0 mmol, 5.0 equiv). Stirring was continued for an additional 5 min before the addition of olefin **1** (0.20 mmol, 1.0 equiv) and bromoalkyne **2** (0.30 mmol, 1.5 equiv). The tube was sealed with a teflon-lined screw cap, removed from the glove box and the reaction was stirred at 25 °C for up to 16 h (the mixture was stirred at 1000 rpm). After the reaction was complete, the reaction was quenched upon the addition of H₂O, and the mixture was extracted with EtOAc. The organic layer was concentrated to give the crude product. *n*-Dodecane (20 μL) was added as an internal standard for GC analysis. The product was purified by flash column chromatography (petroleum ether/EtOAc) for each substrate. See Supplementary Information for more detailed experimental procedures and characterization data for all products.

**Enantioselective NiH-catalyzed hydroalkynylation of styenes**. In a nitrogen-filled glove box, to an oven-dried 8 mL screw-cap vial equipped with a magnetic stir bar were added NiI₂·xH₂O (3.8 mg, 5.0 mol%), (*S*)-**L**\* (3.3 mg, 6.0 mol%), K₃PO₄·H₂O (115.1 mg, 2.5 equiv), NaI (60.0 mg, 2.0 equiv) and anhydrous PhCF₃ (1.0 mL). The mixture was stirred for 20 min at room temperature (stirred at 800 rpm) before the addition of (MeO)₃SiH (64 μL, 0.50 mmol, 2.5 equiv). Stirring was continued for an additional 5 min before the addition of olefin **4** (0.20 mmol, 1.0 equiv) and bromoalkyne **2** (0.30 mmol, 1.5 equiv). The tube was sealed with a teflon-lined screw cap, removed from the glove box and the reaction was stirred at 0 °C for up to 12 h (the mixture was stirred at 800 rpm). After the reaction was complete, the reaction was quenched upon the addition of H₂O, and the mixture was extracted with EtOAc. The organic layer was concentrated to give the crude product. *n*-Dodecane (20 μL) was added as an internal standard for GC analysis. The product was purified by flash column chromatography (petroleum ether/ EtOAc) for each substrate. The enantiomeric excesses (% ee) were determined by HPLC analysis using chiral stationary phases. See Supplementary Information for more detailed experimental procedures and characterization data for all products.

## Data availability

The authors declare that the main data supporting the findings of this study, including experimental procedures and compound characterization, are available within the article and its supplementary information files, or from the corresponding author upon reasonable request.

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

## Acknowledgements

Research reported in this publication was supported by NSFC (21822105, 21772087), NSF of Jiangsu Province (BK20190281, BK20201245), Six Kinds of Talents Project of Jiangsu Province (JNHB-003), programs for high-level entrepreneurial and innovative talents introduction of Jiangsu Province (group program), and Fundamental Research Funds for the Central Universities (020514380263).

## Author contributions

X.J., Y.W., and S.Z. designed the project. X.J., B.H., Y.X., M.D., Z.G., Y.W., and S.Z. co-wrote the manuscript, analyzed the data, discussed the results, and commented on the manuscript. X.J., B.H., Y.X., M.D., and Z.G. performed the experiments. All authors contributed to discussions.

## Competing interests

The authors declare the following competing interest(s): a patent for the synthesis of AMG 837 using this method has been filed.
