## [Peer Review File · Nature Communications]

REVIEWER COMMENTS

Reviewer #1 (Remarks to the Author):

Zhu and co-workers reported on the Ni-catalyzed benzylic alkynylation of styrenes or isolated olefins in high yields and excellent chemo- and regioselectivity. The method allows to forge a new C(sp³)-C(sp) bond at the benzylic positions via the migratory insertion of an olefin into a Ni-H bond followed by a chain-walking mechanism (if starting from isolated olefins) and terminated by the reductive elimination from a benzyl-Ni-alkynyl intermediate. Additionally, an outstanding chemoselectivity for the migratory insertion of the alkenes rather than the alkynes is achieved. The enantioselective hydro-alkynylation of styrenes is also described with chiral benzylic alkynes obtained in good-to-high yields and enantiomeric excesses.

The method reported by the authors represents the first non-photocatalytic strategy for the hydro-alkynylation of olefins and it widens the plethora of chiral benzylic alkynes which can be accessed beyond the use of strongly oxidants (e.g. iodonium salts) or radical-based reagents. The scope of the present methodology is very broad and it showcases the functional group compatibility of the system, due to the very mild reaction conditions employed. That being set, it is unquestionably that this paper will serve as a very powerful endeavor to incorporate the versatile alkynyl motif at benzylic sites from very simple precursors and with high degree of enantioselectivity.

However, the methodology seems to be restricted to a rather particular type of alkynyl halide counterpart, particularly TIPS-protected analogues and only very few variations have been described (2 additional examples in the scope of terminal olefins and 3 more in the enantioselective version with styrenes). It would be insightful to know what is the outcome when alkyl- or aryl-substituted alkynes are employed as coupling partner. In this regard, others have reported that the change in base and ligand allows for the coupling of alkynyl halides other than TIPS-protected ones (see for example, ChemComm 2013, 49, 4286).

Although the authors aim at the development of a remote functionalization strategy through a chain-walking mechanism, there is no example of substrates having the starting C=C bond and the aryl group at more than 2 methylene groups distance. Therefore, the authors are encouraged to include some of these molecules (or a comment throughout the text) in order to understand the limitations and the efficiency of the process.

Noteworthy, the current system is also applicable to internal alkenes substrates and the authors stated that it is insensitive to the stereochemical purity of the starting olefins. However, the initial E/Z ratio is not provided, as well as the one of the unreacted starting material. These information needs to be added since it will support/rule out the intermediacy of a stereospecific process or of a E/Z equilibration preceding the chain-walking event.

Overall, the research has been very well conducted and the results are certainly interesting. I have absolutely no doubt that it will attract considerable interest at the Community. Given the interest in the synthesis of benzylic alkynes, particularly in an enantioselective manner, and the unprecedented strategy adopted, I do recommend acceptance of the manuscript in Nature Communications pending the modifications described above.

Reviewer #2 (Remarks to the Author):

Zhu and Wang reported an Nickel-catalyzed hydroalkynylation of alkenes. As author mentioned, the reaction was initiated by a hydrometallation of alkenes, followed by metal migration on the carbon chain to form benzylic metal complex. Then the benzylic metal complex reacts with alkynyl bromides via single electron transfer process, providing a variety of alkyne products with good to excellent yields. In addition, the asymmetric alkynylation of alkenes was also achieved by introducing chiral pyridinyl-oxazoline ligands, and excellent enantioselectivities was obtained. Moreover, the mechanistic investigation revealed the coupling of benzylic radical with alkynyl-Nickel species. Overall, this is a nice advance in the field of alkene functionalizations, and this

referee recommends to publication in Nature Communication with minor revision.

Comments:

(1) The reaction was conducted with large excess of hydrosilane, which could result the over-reduction of alkene, but this information did not provide. It is better to have these information, which is help reader to understand the chemistry.

(2) For the asymmetric alkynylation, only styrene type substrates was investigated. How about the reactions of terminal alkenes as presented in the top of Figure 3.

(3) Owing to the utility of alkyl-substituted alkynes, how about the reactions with normal alkyl-substituted alkynyl bromides, especial in the case of asymmetric reactions.

Reviewer #3 (Remarks to the Author):

In this manuscript, Zhu et al. developed a nickel-catalyzed migratory hydroalkynylation of alkenes and the enantioselective variant for styrenes. This method features broad substrate scope and good functionality tolerance. Mechanistically, it is a good incremental contribution to the authors' own works [enantioselective hydroarylation: ref. 48, *Angew. Chem. Int. Ed.* 59, 21530–21534 (2020) and enantioselective hydroalkenylation: Ref. 50, *Angew. Chem. Int. Ed.* 60, 4060–4064 (2021)]. In this context, alkynyl bromides were utilized instead of aryl and alkenyl halides. The products of this reaction incorporate a synthetically useful alkynyl moiety, which is the major significance of this work from the viewpoint of organic synthesis. The main issue of this manuscript lies in the introduction part: the authors totally ignore hydroalkynylation of alkenes using terminal alkynes [e. g. *J. Am. Chem. Soc.* 131, 5060 (2009); *Angew. Chem. Int. Ed.* 49, 3827 (2010), *Org. Lett.* 11, 523 (2009) and many other contributions cited in the review *Chin. J. Org. Chem.* 40, 1087 (2020)], which is obviously advantageous over the authors' work from the perspective of both step- and atom-economy, since terminal alkynes can be directly employed as the precursors without prefunctionalization. The authors need to add the discussion of these works to the introduction. This reviewer believes that this paper is a borderline case for publication in *Nat. Commun.*

In response to **reviewer 1** (quotes from reviewer are italicized):

Reviewer #1 (Remarks to the Author):

Zhu and co-workers reported on the Ni-catalyzed benzylic alkynylation of styrenes or isolated olefins in high yields and excellent chemo- and regioselectivity. The method allows to forge a new C(sp³)-C(sp) bond at the benzylic positions via the migratory insertion of an olefin into a Ni-H bond followed by a chain-walking mechanism (if starting from isolated olefins) and terminated by the reductive elimination from a benzyl-Ni-alkynyl intermediate. Additionally, an outstanding chemoselectivity for the migratory insertion of the alkenes rather than the alkynes is achieved. The enantioselective hydro-alkynylation of styrenes is also described with chiral benzylic alkynes obtained in good-to-high yields and enantiomeric excesses.

The method reported by the authors represents the first non-photocatalytic strategy for the hydro-alkynylation of olefins and it widens the plethora of chiral benzylic alkynes which can be accessed beyond the use of strongly oxidants (e.g. iodonium salts) or radical-based reagents. The scope of the present methodology is very broad and it showcases the functional group compatibility of the system, due to the very mild reaction conditions employed. That being set, it is unquestionably that this paper will serve as a very powerful endeavor to incorporate the versatile alkynyl motif at benzylic sites from very simple precursors and with high degree of enantioselectivity.

However, the methodology seems to be restricted to a rather particular type of alkynyl halide counterpart, particularly TIPS-protected analogues and only very few variations have been described (2 additional examples in the scope of terminal olefins and 3 more in the enantioselective version with styrenes). It would be insightful to know what is the outcome when alkyl- or aryl-substituted alkynes are employed as coupling partner. In this regard, others have reported that the change in base and ligand allows for the coupling of alkynyl halides other than TIPS-protected ones (see for example, ChemComm 2013, 49, 4286).

The less steric hindered alkyl substituted ethynyl bromides and aryl substituted ethynyl bromides were unsuccessful substrates. In accord with this comment, as well as a comment from reviewer 2, we have now added a sentence to address this question: "However, it should also be noted that the less steric hindered alkyl substituted ethynyl bromide (**2l'**) and aryl substituted ethynyl bromide (**2m'**) were unsuccessful substrates⁷⁰ and could easily undergo decomposition under current conditions.". We have also cited the above-mentioned paper (see:

ref. 70).

Although the authors aim at the development of a remote functionalization strategy through a chain-walking mechanism, there is no example of substrates having the starting C=C bond and the aryl group at more than 2 methylene groups distance. Therefore, the authors are encouraged to include some of these molecules (or a comment throughout the text) in order to understand the limitations and the efficiency of the process.

We have now added such an example (see: **3l** in Fig. 3) and a sentence to address this question: "The reaction could also proceed with olefin substrate having longer chain length between the starting C=C bond and the remote aryl group, producing the benzylic alkynylation product exclusively although with a lower yield (**3l**)."

Noteworthy, the current system is also applicable to internal alkenes substrates and the authors stated that it is insensitive to the stereochemical purity of the starting olefins. However, the initial E/Z ratio is not provided, as well as the one of the unreacted starting material. These information needs to be added since it will support/rule out the intermediacy of a stereospecific process or of a E/Z equilibration preceding the chain-walking event.

We have now added this information in Fig. 3 and Fig. 4.

Overall, the research has been very well conducted and the results are certainly interesting. I have absolutely no doubt that it will attract considerable interest at the Community. Given the interest in the synthesis of benzylic alkynes, particularly in an enantioselective manner, and the unprecedented strategy adopted, I do recommend acceptance of the manuscript in Nature Communications pending the modifications described above.

In response to **reviewer 2** (quotes from reviewer are italicized):

Reviewer #2 (Remarks to the Author):

Zhu and Wang reported an Nickel-catalyzed hydroalkynylation of alkenes. As author mentioned, the reaction was initiated by a hydrometallation of alkenes, followed by metal migration on the carbon chain to form benzylic metal complex. Then the benzylic metal complex reacts with alkynyl bromides via single electron transfer process, providing a variety of alkyne products with good to excellent yields. In addition, the asymmetric alkynylation of alkenes was also achieved by introducing chiral pyridinyl-oxazoline ligands, and excellent enantioselectivities was obtained. Moreover, the mechanistic investigation revealed the coupling of benzylic radical with alkynyl-Nickel species. Overall, this is a nice advance in the field of alkene functionalizations, and this referee recommends to publication in Nature Communication with minor revision.

Comments:

(1) The reaction was conducted with large excess of hydrosilane, which could result the over-reduction of alkene, but this information did not provide. It is better to have these information, which is help reader to understand the chemistry.

Large excess of hydrosilane did not result the over reduction of alkene. As shown in in Fig. 2, entry 7, similar yield was obtained when reducing the amount of PMHS to 2.5 equiv. We have now added a sentence to address this question: "marginally lower yield was obtained when reducing the amount of PMHS to 2.5 equiv (entry 7)."

(2) For the asymmetric alkynylation, only styrene type substrates was investigated. How about the reactions of terminal alkenes as presented in the top of Figure 3.

Currently, the remote alkenes were not suitable substrates to undergo migratory asymmetric hydroalkynylation. As we already mentioned in Fig. 5a, although the ee is good, the rr is hard to well-control. We have now also added the result of 4-phenyl-1-butene (**1a**) in SI

(see page S73 in SI). Both the yield and rr were poor in this case.

(3) Owing to the utility of alkyl-substituted alkynes, how about the reactions with normal alkyl-substituted alkynyl bromides, especial in the case of asymmetric reactions.

The normal alkyl substituted alkynyl bromides were unsuccessful substrates. In accord with this comment, as well as a comment from reviewer 1, we have now added a sentence to address this question: "However, it should also be noted that the less steric hindered alkyl substituted ethynyl bromide (**2l'**) and aryl substituted ethynyl bromide (**2m'**) were unsuccessful substrates⁷⁰ and could easily undergo decomposition under current conditions."

In response to **reviewer 3** (quotes from reviewer are italicized):

Reviewer #3 (Remarks to the Author):

In this manuscript, Zhu et al. developed a nickel-catalyzed migratory hydroalkynylation of alkenes and the enantioselective variant for styrenes. This method features broad substrate scope and good functionality tolerance. Mechanistically, it is a good incremental contribution to the authors' own works [enantioselective hydroarylation: ref. 48, Angew. Chem. Int. Ed. 59, 21530–21534 (2020) and enantioselective hydroalkenylation: Ref. 50, Angew. Chem. Int. Ed. 60, 4060–4064 (2021)]. In this context, alkynyl bromides were utilized instead of aryl and alkenyl halides. The products of this reaction incorporate a synthetically useful alkynyl moiety, which is the major significance of this work from the viewpoint of organic synthesis. The main issue of this manuscript lies in the introduction part: the authors totally ignore hydroalkynylation of alkenes using terminal alkynes [e. g. J. Am. Chem. Soc. 131, 5060 (2009); Angew. Chem. Int. Ed. 49, 3827 (2010), Org. Lett. 11, 523 (2009) and many other contributions cited in the review Chin. J. Org. Chem. 40, 1087 (2020)], which is obviously advantageous over the authors' work from the perspective of both step- and atom-economy, since terminal alkynes can be directly employed as the precursors without prefunctionalization. The authors need to add the discussion of these works to the introduction. This reviewer believes that this paper is a borderline case for publication in Nat. Commun.

We have now cited all the above-mentioned papers (see: refs. 10-12, and 16). We have modified the introduction paragraph to be more informative, and we have also added a sentence regarding the pioneering works of Suginome.

REVIEWERS' COMMENTS

Reviewer #1 (Remarks to the Author):

The authors have submitted a revised manuscript that has addressed all the minor issues posed by the referees. The cover letter includes a point-by-point response that meets every single point raised by all referees. That being set, the paper can be published as is, no further changes are required.

Reviewer #2 (Remarks to the Author):

Authors have well addressed all the questions raised by this referee. Thus, this manuscript was recommended to publication as it.